# Urban Intellectual Property Strategy and University Innovation: A Quasi-Natural Experiment Based on the Intellectual Property Model City of China

Qinqin Chen , Xingneng Xia, Yuji Hui and Sheng Zhang *

School of Public Policy and Administration, Xi'an Jiaotong University, Xi'an 710049, China; qq0426@stu.xjtu.edu.cn (Q.C.); xiaxingneng1991@stu.xjtu.edu.cn (X.X.); huiyj@stu.xjtu.edu.cn (Y.H.)
* Correspondence: zhangsheng@xjtu.edu.cn

**Abstract:** Colleges and universities play a crucial role in fostering innovation, making it essential to explore effective strategies for promoting innovation at the institutional policy level. This paper focuses on the establishment of intellectual property model cities as a starting point and conducts an empirical analysis using innovation data from 234 cities and 942 colleges and universities between 2007 and 2017. By constructing a multi-temporal double-difference model, this study reveals that the establishment of intellectual property model cities effectively fosters innovation in colleges and universities. Further analysis demonstrates that this promotional effect is particularly significant in the western region, key cities, and key colleges and universities, as well as in the fields of invention and utility model patents. These conclusions withstand a series of robustness tests, confirming their validity. This study reveals that the national intellectual property pilot city policy has a significant influence on university innovation. It achieves this by encouraging investment in research and development and enhancing collaboration in innovation. The findings of this study provide important policy suggestions for maximizing the innovation potential of the intellectual property model city policy. This, in turn, can contribute to economic transformation, upgrading, and the promotion of innovation development in China.

**Keywords:** IP strategy; national IP model city; university innovation; DID model



## 1. Introduction

The national innovation system places great importance on the relationship between government and universities. Universities, as one of the main components of the 'triple helix', are increasingly becoming central institutions in modern society and a crucial driver of knowledge-driven growth [1,2]. Innovation plays a crucial role in driving development, with the protection of intellectual property (IP) being essential for safeguarding innovation [3,4]. The protection of intellectual property is a crucial institutional arrangement for promoting innovation. It serves to safeguard the monopoly interests of inventors and address the positive externalities of innovation [5,6]. In China, governments at all levels have made efforts to enhance intellectual property protection, including the establishment of intellectual property pilots at the city level, aiming to address the existing weaknesses in this area. The national IP pilot cities have undergone six rounds of selection since 2012, encompassing 77 cities across 24 provinces. As a pioneering zone in the country for promoting the construction of the intellectual property system, can the establishment of model cities truly serve as a significant catalyst for fostering innovation?

Institutional policies play a crucial role in facilitating active innovation activities in universities [7]. Universities are instrumental in fostering scientific and technological achievements that bridge the gap between laboratories and markets [8]. These achievements are given high priority by the policies of pilot cities for IP. For instance, the pilot policy emphasizes the promotion of high-value patent cultivation centers in universities

and research institutes. The goal is to achieve a coordinated match between the growth of patent applications, the economic growth rate, and the level of scientific and technological innovation. The policy also aims to facilitate the entire IP process, including layout and design, cultivation and incubation, trading and circulation, and transformation and implementation. Furthermore, it aims to promote IP operation and industrialization in key industries. National IP pilot cities have been implemented for several years, and their construction scope and influence have been continuously expanding. They have become a significant regional innovation policy pilot that cannot be ignored when considering the level of urban innovation. Therefore, as an important progressive reform policy in China's implementation of its innovation-driven development strategy, it is important to investigate whether the innovation-driven policy of national IP pilot cities effectively promotes innovation in universities. And understanding the influence mechanism behind this policy and exploring potential differences in the innovation effects of universities in different cities are crucial. However, these questions have yet to be answered by theoretical frameworks or empirical evidence.

There is a general consensus among academics that moderate IP protection can significantly promote innovation. IP protection for enterprises can discourage technology imitators from infringing, encourage enterprises to engage in research and development (R&D) activities, grant enterprises exclusive rights to innovation results, reduce the spillover of R&D knowledge, and alleviate financing constraints for enterprises [9,10]. That is to say, it can better safeguard the legitimate benefits of enterprise innovation results and improve innovation efficiency [11]. Previous studies have examined the economic and innovation effects of IP model city policy, focusing on dimensions such as urban innovation quality [12], industrial structure upgrade [13], and enterprise innovation level [14]. However, research on IP protection has dominantly concentrated on regional, industrial, and enterprise levels, leaving limited investigation at the university level. The implementation of the Bayh–Dole Act in the United States in 1980 played a crucial role in enhancing innovation activities in universities by encouraging the decentralization of IP [15,16]. Although studies have confirmed the influence of IP on university innovation, most of them have relied on theoretical frameworks and questionnaire survey data, lacking comprehensive data from universities at a micro-level and observations of IP policy interventions.

In response to the limitations of existing research, the following are the innovations of this article: ① For the first time, focusing on the university perspective rather than the traditional enterprise perspective, we utilized the national IP pilot cities as a quasi-natural experiment to assess the impact of establishing IP model cities on university innovation based on innovation data from 942 Chinese universities in 234 prefecture-level cities between 2007 and 2017. It offers a broad sample of data on the influence of IP on university innovation and enriches the quantitative studies on IP policy evaluation. ② We analyzed and validated the role and mechanism of the IP pilot city policy in inducing university innovation. We approached this from two perspectives: promoting R&D investment and strengthening innovation cooperation. Additionally, we explored the pathways through which the pilot policy operates, aiming to provide empirical evidence at the micro-level. This research contributes to the understanding of the influence of IP on university innovation. ③ It examines the heterogeneity of the effect of IP pilot cities on university innovation from four different aspects, city-level, regional, university-level, and patent-type heterogeneity, which provides useful suggestions for identifying the primary element of university innovation and policy action direction.

The remainder of this paper is structured as follows. Section 2 covers the theoretical examination and development of hypotheses. Section 3 presents the research design. The empirical results and analysis are described in Section 4. Section 5 provides the conclusions and suggests policy implications.

## 2. Institutional Background and Theoretical Mechanism

*2.1. Institutional Background*

The Chinese government has consistently prioritized the protection of intellectual property. This commitment was explicitly stated in the 16th National Congress of the Communist Party of China (CPC) in November 2002, which called for improvements in the intellectual property protection system. The 17th National Congress of the CPC in November 2007 further emphasized the importance of implementing a strategy for intellectual property rights as part of the country's innovation agenda. In 2008, the State Council issued the Outline of National IP Strategy, marking the beginning of pilot programs at the city level. And then, Wuhan and 23 other cities were selected as pilot cities in 2012, with the primary goal of promoting innovation-driven development by enhancing the creation, protection, and utilization of IP. This policy covers various aspects, including improving patent quality, ensuring robust IP protection, expediting the development of IP operation and service systems, and continuously innovating IP financial services. It involves multiple stakeholders, such as the government, industry, academia, research, and financial services, forming a comprehensive policy framework to foster innovation and development. The construction of IP pilot cities is a strategic initiative authorized by the central government and implemented by local authorities to strengthen the legal system and stimulate innovation. These cities serve as 'pioneer zones' and 'experimental fields' for exploring and enhancing IP protection. Once the pilot areas have achieved positive policy outcomes and have received recognition from higher levels of government, the aforementioned 'typical experiences' will be disseminated to other cities. This highlights the significance of establishing IP pilot cities as a crucial component of effectively implementing the national IP strategy and fostering the development of a robust IP nation.

*2.2. Theoretical Mechanism*

(1) IP City Policy and University Innovation

The pilot policy on IP cities is a system that aims to enhance China's capacity for independent innovation through targeted, integrated, and dynamic approaches. It focuses on the development and use of knowledge resources, with a specific emphasis on IP. The objective of this policy is to improve the city's ability to create, apply, and protect IP, thereby promoting regional knowledge innovation and contributing to the overall quality and efficiency of the regional economy. It is important to note that the pilot policy on IP cities is distinct from policies implemented at other city levels, such as the low-carbon city policy, which aims to promote overall low-carbon development through energy efficiency, improvements in energy structure, and transformations in the energy industry [17], and the smart city policy, which utilizes information technology to transform urban governance, enabling intelligent urban management, services, and lifestyles [18]. Another crucial requirement for establishing IP pilot cities is to assign a strategic role to IP work in urban development. This entails integrating IP work into the broader context of urban economic and social development, supporting the creation of a favorable environment for IP pilot cities, and generating new opportunities for upgrading the urban industrial structure. Additionally, IP pilot cities are overseen by the State IP Office, which implements a three-level assessment and management mechanism connecting the state, provinces, and municipalities. If a city fails to meet the review standards within three years, it will lose its IP pilot city status.

The establishment of IP model cities can serve as a valuable strategy to mitigate innovation externalities through government funding. Basic research often yields positive knowledge externalities [19,20], which can be enhanced by a robust property rights system that incentivizes innovation behavior [21,22]. The construction of IP pilot cities aims to improve IP administration, enhance IP protection, and provide great convenience for the creation, application, and protection of IP in universities. This, in turn, ensures that technological innovation can lead to economic benefits such as patent authorization and technology transfer. These supportive IP policies can incentivize universities to engage

in more research and development, thereby promoting knowledge innovation. Based on these arguments, this paper proposes the following hypothesis:

**H1:** *The implementation of IP model cities positively impacts university innovation.*

(2)    R&D Investment Intensity and University Innovation

Solow (1956) clearly pointed out that technological progress drives 87.5 percent of economic growth [23]. R&D investment, as a crucial measure and driver of technological innovation activities, plays a significant role in promoting economic growth. It effectively stimulates enthusiasm for R&D and innovation subjects [24]. Resource dependence is a key characteristic of universities, and the impact of R&D innovation, where inputs determine outputs, is particularly evident. R&D inputs typically include human capital and physical capital, with the latter mainly sourced from the government and market. It has been established that when university R&D input exceeds a certain threshold, it can drive the improvement of R&D quantity and quality. Therefore, reasonable university R&D input can effectively enhance innovation output, and a moderate increase in university R&D input is beneficial for achieving the scale effect of innovation output [25,26]. From a macro perspective, there remains a notable disparity between China and other developed countries in terms of R&D intensity and investment structure. China's R&D intensity currently lags behind the level of the United States fifteen years ago, and the proportion of R&D investment in colleges and universities is also significantly lower [27].

At the governmental level, establishing an IP pilot city enhances the government's strategic leadership in IP protection and R&D innovation. To ensure that the construction process of an IP pilot city aligns with the city's innovation system, the local government will optimize the environment for R&D inputs from universities. They will actively increase support for innovation resources, give priority to scientific research funding for universities, and encourage enterprises and social capital to contribute to university R&D. By leveraging the institutional advantages of IP, urban R&D and innovation activities can be revitalized [28]. At the enterprise level, the establishment of IP pilot cities will prompt local governments to increase funding for enterprises during the patent application and authorization process. This will effectively reduce the cost of patenting and R&D innovation, enabling enterprises to expand their R&D investment and further encourage them to invest more scientific and technological funds in universities as their main research base [29]. And at the university level, the policy of IP pilot city improves the institutional environment, facilitating the utilization of scientific and technological advancements by the main innovation entities. Consequently, this encourages the main innovation entities to invest more in innovation due to increased willingness and motivation. Furthermore, the policy of IP pilot city acknowledges and values innovative talents, thereby attracting a greater number of high-caliber individuals and creating a 'public pool' effect. This effect benefits universities and colleges by enabling them to employ more top-notch scientific research talents, resulting in more effective research work [30]. Based on these observations, this paper proposes the following hypothesis:

**H2:** *The pilot policy of IP cities promotes university innovation by enhancing universities' R&D investment.*

(3)    Innovation Cooperation Intensity and University Innovation

Innovation cooperation refers to the exchange, flow, and diffusion of various types of innovation resources to improve the efficiency of innovation by optimizing the allocation of innovation resources [31]. In cities, innovation cooperation primarily depends on the collaboration between enterprises, colleges and universities, and R&D institutions, leveraging their talent pool and educational resources. The cooperation among these entities in the innovation chain is vital for fostering innovation [32]. The partnership between universities and enterprises in innovation cooperation offers regional advantages

and can yield a synergistic effect where the whole is greater than the sum of its parts (1 + 1 > 2). Moreover, universities are more likely to demonstrate higher enthusiasm in engaging in innovation activities such as technology research and development, talent training, and resource sharing with enterprises if there are no IP issues [33]. To meet the major strategic needs of the country and solve technical difficulties, close cooperation between universities, research institutes, and enterprises is necessary. This cooperation should aim to deepen the integration of industry and education and form a mutually beneficial symbiotic innovation ecology.

IP pilot cities can enhance their innovation ecosystem by implementing more favorable innovation policies. These policies can foster collaboration among various innovation stakeholders and promote innovative activities. Local governments can achieve this by formulating relevant policies, providing special funds, establishing cooperation platforms, and organizing joint research projects. These efforts aim to integrate industry, academia, and research into a cohesive innovation system. Furthermore, they facilitate the efficient exchange of innovation factors and strengthen the linkages between universities, localities, industries, and enterprises. This closer collaboration enables universities to effectively bridge the gap between their innovative technologies and the market [34]. To further enhance the development of IP pilot cities, local governments also play a crucial role in leading and guiding universities' IP work. They support the establishment of IP transformation centers, trading centers, and other market-oriented platforms to increase the market value of patents. Additionally, universities are encouraged to set up technology transfer offices to enhance their IP management capabilities [35] and foster collaboration with external partners. This collaboration allows universities to access advanced research equipment, technological platforms, and other resources, facilitating resource sharing, complementary advantages, and the generation of more knowledge outcomes. Based on these observations, this paper proposes the following hypothesis:

**H3:** *The pilot policy of IP cities promotes university innovation by strengthening R&D cooperation.*

### 3. Research Design

*3.1. Double Time-Varying DID Model*

This paper utilizes a fixed-effect double-difference method to examine the influence of the establishment of IP pilot cities on university innovation [36,37]. Out of the 234 cities sampled, a total of 64 cities were approved as IP pilot cities between 2007 and 2017, providing a suitable quasi-natural experiment. The experimental group consists of colleges and universities located in the 64 selected IP pilot cities, while the control group comprises colleges and universities in cities that were not selected. By comparing the experimental and control groups, this study aims to determine the net effect of the national IP pilot city policy on university innovation. To account for the variations in the timing of cities obtaining the national IP pilot city title, the 'asymptotic double-difference method' used in the studies of Beck et al. (2010) and Wang et al. (2019) is adopted in this study to identify the policy effect and test Hypothesis 1 [38,39], as shown in Equation (1).

$$
\begin{aligned}
Univpat_{i,s,t} = {} & \alpha_0 + \alpha_1 Treat_i * Time_{i,s,t} + \alpha_2 Ctrl_{i,s,t} + \beta_i + \gamma_s \\
& + \delta_t + \varepsilon_{i,s,t}
\end{aligned}
\tag{1}
$$

In this model, *i* denotes city, *s* denotes university, and *t* denotes time. *Univpat* denotes university innovation level. The explanatory variable *Treat* ∗ *Time* is the double-difference estimator. *Ctrl* denotes the control variable. $\beta_i$ is the fixed city effect. $\gamma_s$ is the fixed university effect. $\delta_t$ is the fixed time effect. $\varepsilon$ denotes the random disturbance term. The coefficient $\alpha_1$ indicates the policy implementation effect of the national IP pilot city on the impact of innovation in colleges and universities. If $\alpha_1$ is greater than 0, it means that the IP pilot city policy can promote innovation in colleges and universities.

### 3.2. Variable Selection

Explained variables. The explanatory variable in this paper is university innovation, which is measured using the number of patents granted by colleges and universities in the current year (GTtgrapat) [40,41]. This measurement is based on the mainstream practice of existing domestic and international studies. Additionally, the robustness test includes the total number of patent applications (GTtapppat) and the total number of papers published (Paper) by universities in the current year to verify the reliability of the regression results from the benchmark analysis [42,43].

Explanatory variables. The core explanatory variable of this paper is the IP model city (*Treat* ∗ *Time*). Following Yuan et al. (2018) and Hu et al. (2021), it is represented as a dummy variable for IP model cities [44,45]. The variable *Time* is used to measure the impact of policies related to IP pilot cities on the pilot cities. If city *i* is recognized as a national IP pilot city in year t, the variable *Treat* is assigned a value of 1, and the *Time* variable in the subsequent years is also assigned a value of 1. Conversely, if *Treat* is assigned a value of 0, the *Time* variable in the previous years is also assigned a value of 0. The estimation of the cross-multiplier term *Treat* ∗ *Time* coefficient represents the innovation effect of universities in the construction of IP model cities.

Control variables. Based on previous research, this paper selects control variables focusing on the university and city levels [46]. At the university level, variables such as the number of appraised achievements of universities (Numir), the amount of scientific and technological funds allocated for the year (Amtastf), the actual income of technology transfer of universities for the year (Rittc), the total number of scientific and technological projects (Tnumsttp), and the number of scientists and engineers (Setrp) in the year were considered [47–49]. At the city level, variables such as the level of financial development (Finadevelop), the level of economic development (PcptlGRP), industrial structure level (Industrlevel), and science and technology expenditure (STspend) in urban municipal districts were taken into account [50–52]. Please refer to Table 1 for detailed definitions of these variables.

**Table 1.** Key variables and definitions.

| Variable | Definitions |
| --- | --- |
| GTtgrapat | The logarithm of the sum of the number of inventions patents, utility model patents, and design patents of university in the year + 1 |
| Treat ∗ Time | Grouping dummy variables multiplied by policy implementation dummy variables, which refers to the innovation effect of universities in intellectual property model cities in this paper |
| Numir | The logarithm of the number of scientific and technological ((S&T)) achievements validated by university in the year + 1 |
| Rittc | The logarithm of the actual income from technology transfer of university in the year + 1 |
| Amtastf | The logarithm of the sum of government transfers, enterprise transfers, and other sources of funding to university in the year + 1 |
| Tnumsttp | The logarithm of the total number of S&T projects of universities in the current year + 1 |
| Setrp | The logarithm of the sum of the number of scientists and engineers in teaching and research staff and R&D staff of university in the year |
| Finadevelop | The ratio of the balance of loans from financial institutions to regional GDP at the end of the year in urban municipal districts |
| PcptlGRP | The regional GDP per capita in urban municipal districts |
| Industrlevel | The share of secondary sector in GDP in urban municipal districts |
| STspend | The logarithm of the amount of science and technology expenditures in urban municipal districts |

*3.3. Data Description*

This study focuses on the data regarding science and technology activities of Chinese universities and the economic data at the corresponding city level of these universities from 2007 to 2017. The S&T and patent data at the university level were primarily obtained from the Compendium of S&T Statistics for Higher Education Institutions and the China Research Data Service Platform (CNRDS) database. The city-level data were sourced from the China Urban Statistical Yearbook of previous years, while the data on IP model cities were collected from the official website of the State IP Office (SIPO). In order to obtain empirical data that meet the requirements of the empirical research in this paper, the raw data obtained through the aforementioned methods were processed as follows: Firstly, city samples that underwent administrative division adjustments during the period of 2007–2017 and had significant missing data on key variables were excluded (such as Hami City, Danzhou City, Linzhi City, Nachu City, Shannan City, etc.) [53]. Secondly, specialized colleges and universities were excluded from the sample, as well as undergraduate colleges and universities with significant missing data on key variables (such as Kashi University, Changsha Medical College, Haikou College of Economics, Yunnan Police College, Southwest University of Political Science and Law, and other colleges and universities). Thirdly, to address the issue of partially missing indexes for the samples of cities, colleges, and universities, we supplemented the missing data by referring to the annual reports of statistics and using linear interpolation. Additionally, in order to mitigate the impact of outliers, all continuous variables were logarithmically transformed and winsorized at the upper and lower 1% levels. Finally, a total of 6668 observations were obtained, including 50 cities in the experimental group, 184 cities in the control group, and 942 general colleges and universities. Table 2 presents the descriptive statistics of the main variables.

**Table 2.** Descriptive statistics.

| Variable | Obs | Mean | Std. Dev. | Min | Max |
|---|---|---|---|---|---|
| lnGTtgrapat | 6668 | 1.132842 | 2.129193 | 0 | 12.54 |
| lnNumir | 5771 | 1.534961 | 1.458188 | 0 | 4.727388 |
| lnAmtastf | 6668 | 10.32499 | 2.163569 | 0 | 14.45748 |
| lnRittc | 6668 | 3.284496 | 3.715637 | 0 | 10.76266 |
| lnTnumsttp | 6668 | 5.528675 | 1.555761 | 0 | 8.495766 |
| lnSetrp | 6668 | 6.799968 | 1.114953 | 3.367296 | 9.422706 |
| Finadevelop | 5771 | 1.311684 | 0.6597962 | 0.3620147 | 3.288189 |
| PcptlGRP | 5771 | 5.994771 | 3.207033 | 1.2032 | 15.0853 |
| Industrlevel | 6636 | 4.442355 | 1.046416 | 1.9265 | 6.8975 |
| lnSTspend | 6668 | 11.51324 | 1.582075 | 8.409608 | 15.04431 |

## 4. Empirical Results and Analyses

*4.1. Benchmark Analysis*

Table 3 presents the results of the baseline model estimates, examining the impact of the IP pilot city policy on university innovation. Models (1), (2), and (3) all include fixed university effects, fixed city effects, and fixed time effects. Model (2) additionally incorporates university-level control variables, while model (3) includes both university-level and city-level control variables. The estimated coefficients for the policy variable 'IP pilot city' consistently show positive values in all models, regardless of the inclusion of control variables. Furthermore, all coefficients pass the significance test at the 1% level, indicating that the IP pilot city policy effectively promotes innovation in colleges and universities within the pilot region. As control variables are added at the university and city levels, the estimated coefficients for the policy variable decrease, suggesting the presence of other factors that influence university innovation at these levels. This emphasizes the importance of considering these factors to obtain a more accurate estimation of the net effect of the policy. Model (3) demonstrates that after accounting for potential interfering factors, the implementation of the IP pilot city policy significantly promotes university

innovation, leading to an approximately 55% increase in the number of patents granted by universities in the pilot region. Consequently, Hypothesis 1 is confirmed.

**Table 3.** Benchmark regression estimates.

| Variables | (1) | (2) | (3) |
|---|---|---|---|
| Treat * Time | 0.699 *** (11.04) | 0.648 *** (10.28) | 0.551 *** (8.687) |
| lnNumir | | 0.0476 ** (2.058) | 0.0519 ** (2.243) |
| lnAmtastf | | 0.00860 (1.176) | 0.00670 (0.921) |
| lnRittc | | −0.265 *** (−7.081) | −0.254 *** (−6.815) |
| lnTnumsttp | | 0.0228 (0.423) | 0.0151 (0.280) |
| lnSetrp | | −0.0348 (−0.290) | −0.0760 (−0.631) |
| Finadevelop | | | −0.0618 (−0.946) |
| PcptlGRP | | | 0.0891 *** (5.066) |
| Industrlevel | | | −0.243 *** (−5.542) |
| InSTspend | | | 0.135 ** (2.264) |
| R-squared | 0.7836 | 0.789 | 0.792 |
| University FE | NO | YES | YES |
| Year FE | NO | YES | YES |
| City FE | NO | YES | YES |
| Observations | 6582 | 5654 | 5646 |

Notes: Robust t-statistics in parentheses; *** $p < 0.01$, ** $p < 0.05$. University FE = fixed university effect. Year FE = fixed year effect. City FE = fixed city effect.

### 4.2. Robustness Test

#### 4.2.1. Tobit Model

To address potential estimation bias, the Tobit model is employed to re-test the benchmark regression results, considering the restricted dependent variable of the number of patents granted by universities in the current year (GTtgrapat). The regression results are presented in Table 4, with column (1) showing the results of the Mixed Logit model, column (2) displaying the results of the Random Effects Logit model, and column (3) presenting the marginal effects of the Random Effects Logit model. All regression coefficients are found to be significantly positive at the 1 percent level, indicating that the IP pilot policy indeed has a significant impact on innovation in universities. These findings align with the robustness of the benchmark regression results reported earlier in this study.

**Table 4.** Tobit model regression results.

| Variables | Mixed Logit Model | Random Effects Logit | Marginal Effects |
|---|---|---|---|
| Treat * Time | 0.7159 *** (11.61) | 0.6608 *** (8.16) | 0.6608 *** (8.16) |
| R-squared | 0.6207 | 0.4723 | |
| Control variables | YES | YES | |
| University FE | YES | YES | |
| Year FE | YES | YES | |
| City FE | YES | YES | |
| Observations | 5763 | 5763 | |

Notes: Robust t-statistics in parentheses in column (1); z-statistics in parentheses in column (2) and (3); *** $p < 0.01$.

#### 4.2.2. Parallel Trend Test

The unbiased estimation results of the multi-period double-difference method depend on the benchmark regression model meeting the parallel trend assumption. This assumption states that colleges and universities in IP pilot cities and non-IP pilot cities should have similar trends of change before policy implementation. Failure to meet this assumption can result in either overestimation or underestimation of the policy's effect. To test the parallel trend, this paper adopts the processing method used by Beck et al. (2010) and presents the test results in Table 5. The base period for the policy is set as the year

before its implementation, with the current period being 2012. The parallel trend test covers data from 5 years before the policy's occurrence to 5 years after its implementation. The coefficients of each policy point are considered significant at a 90% level. Figure 1 shows that the impact of the IP model city policy on colleges and universities in the host city did not pass the significance level test before recognition. This indicates that there was no significant difference in the innovation level of colleges and universities between the model city and the non-model city prior to the assessment, confirming the assumption of the parallel trend. After being recognized as an IP model city, the innovation level of colleges and universities in the model city exhibited an upward trend without any time lag effect. This suggests that the promotion effect of the national IP model city policy is gradually increasing.

**Table 5.** Parallel trend test.

| Variables | (1) | Variables | (2) |
|---|---|---|---|
| pre5 | −0.264 (−1.161) | post1 | 0.364 *** (3.781) |
| pre4 | 0.299 (1.475) | post2 | 0.537 *** (5.397) |
| pre3 | −0.231 (−1.182) | post3 | 0.689 *** (6.319) |
| pre2 | −0.159 (−1.618) | post4 | 0.800 *** (6.724) |
| 0 | 0.182 * (1.924) | post5 | 1.060 *** (7.016) |
| Observations | 6582 | Observations | 6582 |
| R² | 0.786 | R² | 0.786 |

Notes: Robust t-statistics in parentheses; *** $p < 0.01$, * $p < 0.1$.

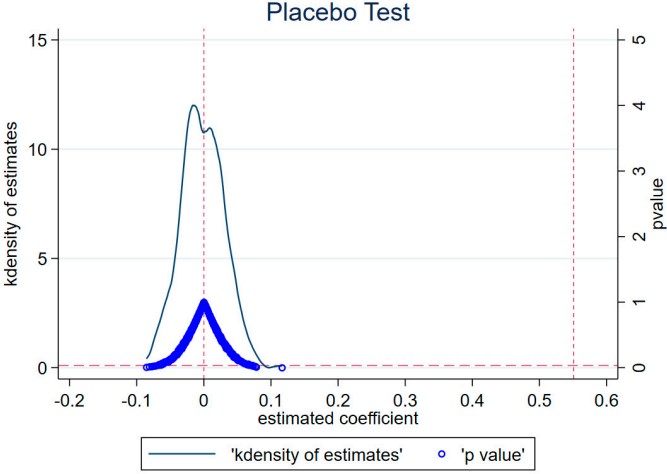

**Figure 1.** IP pilot city policy placebo test.

### 4.2.3. Placebo Test

The placebo test is a valid method to ensure that the baseline regression results are not influenced by random chance events. In this study, we conducted a policy placebo test by randomly selecting the experimental group and estimating the model 500 times through a random simulation process. Figure 1 in this study illustrates the results of the test. The horizontal axis in Figure 1 represents the estimated coefficient of the policy effect, while the vertical axis represents the kernel density value and the $p$-value of the estimated coefficient. Figure 1 demonstrates that the mean value of the estimated coefficients for the policy effect is 0, with most of the $p$-values above 0.1. Moreover, the actual estimated coefficients for the policy effect of the IP pilot city fall within the range of low-probability events in the placebo test plot. Therefore, the impact of the IP pilot city policy on university innovation is not a result of random chance, and the findings of this study are robust and reliable.

### 4.2.4. Propensity Score-Matching–Double-Difference Method (PSM-DID)

As an exogenous policy shock, the IP pilot cities have effectively tackled the issue of endogeneity. Nevertheless, it is crucial to acknowledge that the selection of pilot city areas might not have been random, leading to potential variations between different cities and universities. These variations could introduce some 'noise' to the policy evaluation results in this study. To address this concern, we utilize propensity score matching to identify a comparable control group for each experimental group. We then use the matched samples to estimate the logit model. We initially tested the matching equilibrium hypothesis, and the results are presented in Figure 2. The comparison between the density functions of treated and untreated cities before and after matching equilibrium reveals a significant overlap. However, it is important to note that there is a positive bias in the distribution of cities within the treated and untreated samples, particularly in terms of their level of economic development, financial development, and science and technology expenditures. Figure 3 illustrates the relative magnitude of the variable-specific bias, and the absolute value of the standardized percentage for all variables is less than 20 percent. This indicates that there were no systematic differences between the treated and control groups after matching, validating the results of the matching treatments.

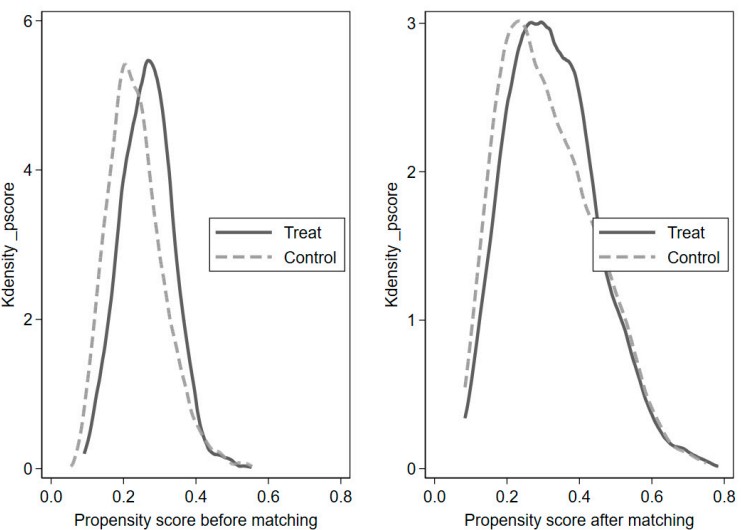

**Figure 2.** Balancing before and after the matching procedure.

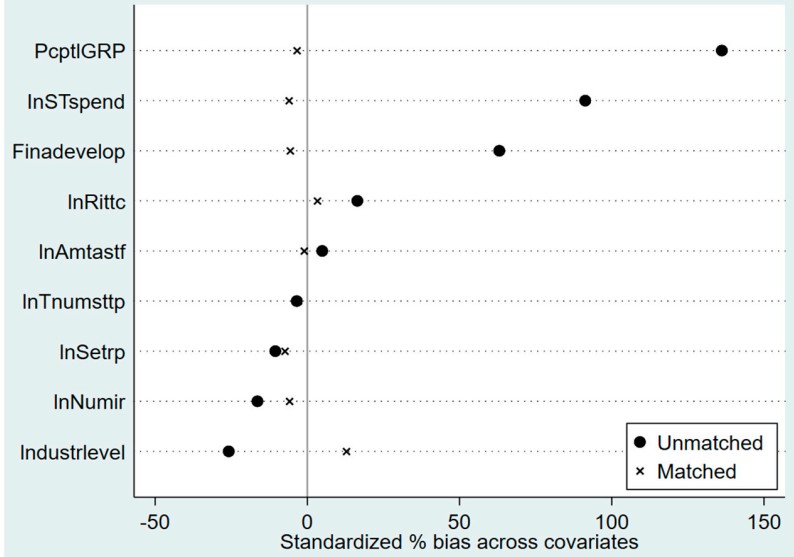

**Figure 3.** Standardized percentage bias across covariates.

After passing the matching equilibrium assumption, we utilized a year-by-year matching method to screen the control group colleges. The model regression estimation results using the PSM-DID method are presented in Table 6, employing three matching methods: k-nearest neighbors, nuclear matching, and radius matching. The results in Table 6 indicate that the coefficient value of the policy effect is significantly positive at the 1% level for all three approaches. This suggests that, even after accounting for differences in university and city characteristics, the IP pilot city policy plays a significant role in promoting innovation in universities. Therefore, the estimation results of PSM-DID further reinforce the robustness of the findings presented in this paper.

**Table 6.** PSM-DID test.

| Variables | K-Nearest Neighbour Matching | Nuclear Matching | Radius Matching |
|---|---|---|---|
| Treat * Time | 0.4588 *** (4.7681) | 0.5728 *** (9.0112) | 0.5783 *** (8.9561) |
| R-squared | 0.7808 | 0.7723 | 0.7735 |
| Control variables | YES | YES | YES |
| University FE | YES | YES | YES |
| Year FE | YES | YES | YES |
| City FE | YES | YES | YES |
| Observations | 5091 | 6075 | 5825 |

Notes: Robust t-statistics in parentheses; *** $p < 0.01$. Control variables = university-level and city-level control variables.

### 4.2.5. Lag Variable Model

To address the potential time lag in innovation activities and the issue of endogeneity between variables and the IP pilot city policy, we have estimated the model with explanatory variables lagged by one and two periods. The results of the regression analysis, shown in columns (1) and (2) of Table 7, reveal that the coefficients on the policy effects are significantly positive. This confirms the reliability and consistency of our baseline model estimates.

**Table 7.** Lagged period and replacement of explanatory variables test.

| Variables | (1) | (2) | (3) | (4) |
|---|---|---|---|---|
| Treat * Time | 0.0826 *** (3.150) | 0.0956 *** (3.477) | 1.242 *** (12.28) | 0.2698 ** (2.16) |
| R-squared | 0.819 | 0.845 | 0.800 | 0.856 |
| Control variables | YES | YES | YES | YES |
| University FE | YES | YES | YES | YES |
| Year FE | YES | YES | YES | YES |
| City FE | YES | YES | YES | YES |
| Observations | 4637 | 3987 | 6582 | 5600 |

Notes: Robust t-statistics in parentheses; *** $p < 0.01$, ** $p < 0.05$.

### 4.2.6. Substitution of Explanatory Variables

To ensure that differences in the measurement of explanatory variables do not affect the model estimation results, we replaced GTtgrapat with GTtapppat and Paper, respectively. The model estimation results, presented in columns (3) and (4) of Table 7, demonstrate that the significance and direction of the estimated coefficients remain unchanged. This further supports the robust impact of the national IP pilot city policy on the promotion of innovation in universities.

### *4.3. Heterogeneity Analysis*

### 4.3.1. City Hierarchical Heterogeneity

Chinese cities are classified into different political levels, which leads to varying allocation of resources such as funds, infrastructure, and preferential policies. This can

result in different roles for IP model cities. In order to address this, the cities belonging to the sample universities are categorized into key city groups and general city groups. Key city groups consist of provincial capital cities, while general city groups include ordinary prefecture-level cities. The regression results in Table 8, columns I and II, report the impact of the IP pilot city policy on the innovation level of universities in cities at different levels. The results indicate that the empirical *p*-value of the between-group coefficient of the two groups of cities is less than 0.01, which means the difference in between-group coefficients is significant. Thus, it is meaningful to directly compare the estimated coefficients of the double-difference terms between cities of different political levels, and the estimated coefficients of the double-difference terms for colleges and universities in the key city group are larger than those in the general city group and both of them are significantly positive. This suggests that the IP pilot city policy has a significant effect on the innovation level of universities in cities at different levels, with a more pronounced effect on universities in the key city group. The comparative advantages enjoyed by key cities in terms of innovation resources, institutional environment, innovation atmosphere, and economic development level contribute to this phenomenon. As a result, colleges and universities in these key cities are more likely to attract innovation resources and stimulate innovation motivation through their IP policies.

**Table 8.** Heterogeneity analysis in different cities.

| Variables | General Cities | Key Cities | Eastern Cities | Central Cities | Western Cities |
|---|---|---|---|---|---|
| Treat $*$ Time | 0.517 *** | 0.779 *** | 0.422 *** | 0.556 *** | 0.801 *** |
|  | (4.687) | (7.989) | (4.114) | (6.313) | (5.355) |
| Control variables | YES | YES | YES | YES | YES |
| R-squared | 0.775 | 0.807 | 0.820 | 0.724 | 0.746 |
| University FE | YES | YES | YES | YES | YES |
| Year FE | YES | YES | YES | YES | YES |
| City FE | YES | YES | YES | YES | YES |
| Observations | 3498 | 2148 | 2638 | 1747 | 1261 |
| Empirical *p*-value | 0.001 *** | | 0.002 *** | 0.047 *** | 0.000 *** |

Notes: Robust t-statistics in parentheses; *** $p < 0.01$. The empirical *p*-value for the test of difference of between-group coefficient analyzed for heterogeneity was calculated using the Fisher's permutation test (2000 samples).

### 4.3.2. City Regional Heterogeneity

As the frontier of reform and opening up, the eastern region has a head start in terms of economic development, a well-established factor market, and better conditions for innovation and development compared to the central and western regions. Therefore, the establishment of IP model cities may have varying effects in the east, central, and western regions due to their locational differences. To investigate this, this paper conducts regression analyses on sample cities in each region to examine the heterogeneity in the innovation effect of universities based on their location. The regression results of the IP pilot city policy on the innovation level of universities in different location cities are presented in columns III, IV, and V of Table 8. The results indicate that the empirical *p*-value of the between-group coefficient between a single group of location cities and the other two groups of location cities all are less than 0.01; that is to say, the difference of between-group coefficients is significant, and the direct comparison of the estimated coefficients of the double-difference terms within different location cities is meaningful. The estimated coefficients of the double-difference terms are the largest for urban universities in the western region, and the smallest in the eastern region, and all of them are significantly positive. This suggests that the IP pilot city policy has a significant effect on the innovation level of universities in different locations, with a more pronounced effect observed in the western city group. This may be attributed to the fact that, compared to eastern cities, western cities are generally more economically underdeveloped, resulting in weaker IP

awareness and lower IP protection capabilities. Consequently, universities in western cities are more strongly motivated by IP policies and have greater potential for improvement.

### 4.3.3. University Grade Heterogeneity

Chinese universities are classified into different administrative levels, which results in differences in the distribution of financial resources, human resources, and research platforms. This differentiation can lead to the establishment of IP model cities with varying effects. In order to address this, the colleges and universities sampled in this study are divided into two groups: key colleges and universities and general colleges and universities. The key colleges and universities include those classified as '211' and co-construction of provincial and subordinate universities, while the general colleges and universities comprise the remaining ordinary institutions. The regression results of the IP pilot city policy on the innovation level of these different groups of universities are presented in columns I and II of Table 9. The results demonstrate that the empirical *p*-value of the between-group coefficient between the two groups of colleges and universities is less than 0.01, which means the difference of between-group coefficients is significant, and the direct comparison of the estimated coefficients of the double-difference terms between the two groups is meaningful. The estimated coefficients of the double-difference terms are significantly positive for both groups of universities, with a larger effect observed for the key universities group. This indicates that the impact of the IP pilot city policy on the innovation level of universities varies depending on their level, with a more noticeable effect on key universities. This can be attributed to the comparative advantages that key universities have in terms of research conditions, human resources, institutional environment, innovation atmosphere, and brand effect. These advantages enable them to optimize the allocation of innovation resources more effectively, leading to more positive outcomes in response to the incentives provided by the IP policy.

**Table 9.** Heterogeneity analysis in different universities.

| Variables | Key University | General University | Invention Patent | Utility Model Patent | Design Patent |
|---|---|---|---|---|---|
| Treat * Time | 0.727 *** | 0.436 *** | 0.253 *** | 0.275 *** | 0.0290 ** |
| | (4.021) | (7.347) | (8.147) | (8.216) | (2.567) |
| Control variables | YES | YES | YES | YES | YES |
| R-squared | 0.848 | 0.679 | 0.841 | 0.673 | 0.553 |
| University FE | YES | YES | YES | YES | YES |
| Year FE | YES | YES | YES | YES | YES |
| City FE | YES | YES | YES | YES | YES |
| Observations | 1009 | 4636 | 5646 | 5646 | 5646 |
| Empirical *p*-value | 0.000 *** | | 0.007 *** | 0.000 *** | 0.000 *** |

Notes: Robust t-statistics in parentheses; *** $p < 0.01$, ** $p < 0.05$.

### 4.3.4. Patent Type Heterogeneity

The State IP Office (SIPO) classifies patents into three types: invention patents, utility model patents, and design patents. The difficulty of examining these patents decreases in the mentioned order. Invention patents are considered to be upstream patents in the chain of innovation activities, focusing more on technological research, development, and process innovation. On the other hand, utility model patents and design patents are seen as downstream patents in the chain of innovation activities, with a greater emphasis on product research, development, and product innovation. The value of different types of patents in universities varies, which can lead to different roles in the establishment of IP model cities. To explore this, this paper conducts regression analyses on invention patents, utility model patents, and design patents of universities to examine the heterogeneity of innovation effects based on patent type. The regression results of the IP pilot city policy on the innovation level of different types of patents in universities are reported in columns III, IV, and V of Table 9. The results indicate that the empirical *p*-values of the coefficients

between groups of single-category patents are all less than 0.01, i.e., the coefficients are significantly different between groups, and a direct comparison of the estimated coefficients of the double-difference terms between groups is meaningful. The estimated coefficients of the double-difference terms are higher for invention patents and utility model patents compared to design patents, and all of them are significantly positive. The implementation of the IP pilot city policy has a significant influence on the level of innovation in various types of patents within universities. It has the ability to enhance both process innovation and product innovation in universities, particularly in promoting high-tech value patents.

*4.4. Mechanism Analysis*

Based on the theoretical analysis discussed in the previous section, the IP pilot city policy primarily fosters innovation in universities by focusing on two action pathways: increasing R&D investment and promoting innovation cooperation. In line with this, this paper utilizes the research conducted by Beck et al. [38] and Yu J et al. [54] to develop recursive equations for evaluating the action mechanisms of universities' innovations. These equations are derived from model (1) and are able to further test Hypothesis 2 and Hypothesis 3, represented by Equations (2) and (3):

$$M_{i,s,t} = {}_0 + {}_1 Treat_i * Time_{i,s,t} + {}_2 Ctrl_{i,s,t} + \beta_i + \gamma_s + \delta_t + \varepsilon_{i,s,t} \tag{2}$$

$$UnivPat_{i,s,t} = \phi_0 + \phi_1 Treat_i * Time_{i,s,t} + \eta M_{i,s,t} + \phi_2 Ctrl_{i,s,t} + \beta_i + \gamma_s + \delta_t + \varepsilon_{i,s,t} \tag{3}$$

In Equation (2), M is the mechanism variable. We use the full-time equivalent research and development personnel (Fterdp) and the amount of internal expenditure on science and technology in the current year of universities (Amtiexpstf) to characterize the R&D investment mechanism variable, the number of patents jointly granted in the current year of universities (Utgrapat) to characterize the innovation cooperation mechanism variable, and the rest of the variables have the same meanings as in Equation (1) [26]. In Equation (3), $\phi_1$ is the regression coefficient of the innovation effect of universities after adding the role and mechanism variable, and $\eta$ is the regression coefficient of the role and mechanism variable.

The results of the mechanism tests are presented in Table 10. In columns (1), (3), and (5), the regression coefficients of Fterdp, Amtiexpstf, and Utgrapat's IP pilot city policies are all significantly positive, suggesting that IP pilot city policies promote R&D investment and innovation cooperation in universities. After controlling for the corresponding mechanism variables, the impact of IP pilot city policies on university innovation is examined in columns (2), (4), and (6). The results show that the regression coefficients of the IP pilot city policy remain statistically significant even after considering the above three role and mechanism variables. However, compared to the benchmark regression coefficients of 0.551 for the IP pilot city policy in column (3) of Table 2, the regression coefficients decrease by 0.05, 0.041, and 0.163, respectively. This finding aligns with the logic of the role and mechanism test. Hence, it can be concluded that promoting R&D investment and enhancing R&D cooperation serve as the mechanisms through which IP model cities can enhance university innovation. Consequently, Hypothesis 2 and Hypothesis 3 are supported.

**Table 10.** The result of mechanism test.

| Variables | Fterdp | GTtgrapat | Amtiexpstf | GTtgrapat | Utgrapat | GTtgrapat |
|---|---|---|---|---|---|---|
| Treat * Time | 0.035 *** | 0.501 *** | 0.021 *** | 0.510 *** | 0.027 *** | 0.388 *** |
| | (2.46) | (8.70) | (2.73) | (8.24) | (5.02) | (7.18) |
| Control variables | YES | YES | YES | YES | YES | YES |
| R-squared | 0.790 | 0.792 | 0.865 | 0.797 | 0.783 | 0.854 |
| University FE | YES | YES | YES | YES | YES | YES |
| Year FE | YES | YES | YES | YES | YES | YES |
| City FE | YES | YES | YES | YES | YES | YES |
| Observations | 5646 | 5646 | 5367 | 5367 | 5646 | 5646 |

Notes: Robust t-statistics in parentheses; *** $p < 0.01$.

## 5. Conclusions and Policy Implications

### 5.1. Conclusions

This paper presents a theoretical analysis of the impact of the construction of IP model cities on the innovation level of Chinese colleges and universities. This study focuses on a sample of 942 undergraduate institutions in 234 prefecture-level cities, spanning the period from 2007 to 2017. By employing a multi-temporal double-difference model, the empirical analysis reveals that the establishment of IP pilot cities has a positive effect on the innovation level of colleges and universities. However, this effect varies across cities and institutions, with higher administrative-grade cities experiencing a stronger enhancement compared to ordinary prefectures. Moreover, colleges and universities in the western region exhibit a stronger enhancement effect than those in the central and eastern regions. Additionally, this study finds that the impact on innovation is stronger for high-administrative-grade institutions compared to other general undergraduate colleges. Furthermore, the analysis indicates that the role of innovation enhancement is greater for invention and utility model patents, as opposed to design patents in colleges and universities. Mechanism tests demonstrate that increased R&D investment and enhanced innovation cooperation serve as effective mechanisms for IP model cities to enhance the innovation level in colleges and universities. These conclusions are robust even after conducting various tests, such as the Tobit model, the parallel trend test, the placebo test, the propensity score-matching–double=difference method test, the lagged period model test, and the replacement of explanatory variables.

### 5.2. Policy Implications

The findings of this study provide valuable insights for the national IP pilot city policy and contribute to the ongoing debate on the impact of IP in protecting innovation. Firstly, policymakers need to have strong confidence in the national IP model city policy and expand the scope of IP model cities, especially among cities in the western region. The establishment of IP pilot cities has already achieved significant success, and policymakers should continue to favor resources to further promote the IP pilot cities policy. Additionally, it is important to explore the broader implications of the IP system and integrate IP strategy into the city's economic and social development, thereby enhancing the strategic position of IP. Meanwhile, because the impact of IP pilot city policies varies among different cities, districts, and universities, policymakers should take into account the actual situation of local economic development and the concentration of universities. On the one hand, they should strengthen policy in pilot cities that can more significantly promote university innovation, and on the other hand, they also should explore in-depth how to better enhance the innovation effect of universities in cities where the role of IP pilot policies is not very significant.

Secondly, to better promote university innovation, policymakers can focus on increasing R&D investment and enhancing R&D cooperation in universities. Although direct government funding for R&D in universities is limited, the government can guide the investment of diversified social capital in university research activities through policy. Additionally, the government can promote cooperation between universities and enterprises by encouraging enterprises to establish joint research institutes with universities, promote the enhancement of research capabilities in universities by supporting universities to lead or participate in the construction of regional laboratories, and accelerate the transfer and industrialization of S&T achievements in universities by launching a special initiative to promote the transfer of S&T achievements. These measures will increase incentives for universities to foster innovation.

Thirdly, policymakers should focus on the synergistic effect of the pilot IP policy with other policies to form a policy synergy to better drive university innovation. While the IP pilot city policy has clearly promoted innovation in universities, it also indicates that other effective policies to foster innovation in universities are insufficient. Studies have demonstrated that policies such as financial assistance and industry–university research



cooperation can also facilitate innovation, which means that these policies can work in conjunction with intellectual property policies by generating a synergistic effect to promote innovation in universities more effectively and efficiently. Policymakers should be bold enough to try to find maximum policy synergies to better stimulate innovation in universities.

### 5.3. Limitations and Directions for Further Research

This paper examines the impact of IP pilot city policies on universities. It contributes to the evaluation of these policies and provides empirical evidence of the innovation effect of IP in universities. Future research can expand on this study in the following ways: Firstly, the indicators used to measure innovation in universities should be expanded. In addition to quantity, focus on quality to obtain more accurate results. Secondly, the sample scope should be extended to include urban policy practices in other developing and developed countries. The applicability of the conclusions to these regions is yet to be verified. Thirdly, the categorization of universities should be refined, and policy research on universities with different orientations, such as teaching orientation and research orientation, should be conducted to obtain more specific research findings.

**Author Contributions:** Conceptualization, S.Z. and Q.C.; methodology, X.X.; software, Q.C.; validation, Y.H. and S.Z.; formal analysis, Q.C.; investigation, Y.H.; resources, X.X.; data curation, Y.H.; writing—original draft preparation, Q.C.; writing—review and editing, Q.C. and S.Z.; visualization, X.X.; supervision, S.Z.; project administration, S.Z.; funding acquisition, S.Z. All authors have read and agreed to the published version of the manuscript.

**Funding:** This research received no external funding.

**Data Availability Statement:** This datasets are available from Q.C. on reasonable request.

**Conflicts of Interest:** The authors declare no conflict of interest.

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
