# Peer review of "Urban Intellectual Property Strategy and University Innovation: A Quasi-Natural Experiment Based on the Intellectual Property Model City of China"

_systems, doi:10.3390/systems12010021_

Round 1
Reviewer 1 Report
Comments and Suggestions for Authors
1. There should be no abbreviations in the abstract.
2. What is the innovation of the paper? Please add this section.
3. Why are there not as many observations when control variables are added to the model?
4. Add a parallel trend test plot.
5. If using the PSM method, it should not be a direct regression. Add a test of balance, as well as a plot of kernel density before and after matching, and a plot of kernel density after matching.
6. Conduct a robustness test using the Tobit model.
7. Please add between-group coefficient test to validate between-group differences of subgroups in the heterogeneity test.
8. H2: The pilot policy of IP city promotes university innovation by enhancing universities' R&D investment. H3: The pilot policy of IP cities promotes university innovation by strengthening R&D cooperation. In the hypothesis section, the authors clearly use R&D investment as well as R&D cooperation as mediating mechanisms. Why does the mechanism analysis section use a triple difference model? This is incorrect. The authors either use a mediation model or rewrite the hypothesis section.
9. The triple difference model is essentially a derivative of the moderating effects model. Moderating effects models should include more than just interaction terms. It does not make economic sense to observe the significance of the coefficients of treat*time*M for the triple difference model only. Please refer to the Moderating Effects Model to modify the Triple Difference Model.
Comments on the Quality of English LanguageMinor editing of English language required.
Author Response
We appreciate the opportunity to revise our paper “Urban Intellectual Property Strategy and University Innovation: A Quasi-Natural Experiment from National Intellectual Property Model City”. Thank you very much for taking the time to review this manuscript. We have taken the suggestions to heart and believe we have successfully addressed all issues. Thus, we now upload a revised version of the manuscript that is now in much better shape after the process. You will also find a complete list of the detailed changes we have made in response to the comments on a point-by-point basis. We appreciate your willingness to review this version of our paper. Please let us know if you need any additional information from us.
Point-by-point response to Comments and Suggestions for Authors
|
Comments 1: There should be no abbreviations in the abstract. |
|
Response 1: Accepted. We accept your opinion and change all abbreviations in the abstract to full names. Please see the abstract on page 1, lines 51 to 65 , in the revised version for details.
|
|
Comments 2: What is the innovation of the paper? Please add this section. |
|
Response 2: Accepted. Your opinion has been considered and the innovations have been itemized more clearly in the penultimate paragraph of the introduction section. Please see the penultimate paragraph on page 2, lines 173 to 188, in the revised version for details.
Comments3: Why are there not as many observations when control variables are added to the model? Response 3: The decrease in sample observations after adding control variables to the model can be attributed to the unbalanced panel data nature of our sample. To maximize the sample size, we processed the raw data in such a way that only samples with severe missing key variables were excluded, and the remaining samples still had a few missing variables in some years. And these missing variables we couldn't fill in by referring to statistical almanacs and using linear interpolation.
Comments4: Add a parallel trend test plot. Response 4: Accepted. Following your opinion, we replace the original parallel trend test table with a parallel trend test plot. Please see the Figure 1 on page 9, lines 1704 to 1705, in the revised version for details.
Comments5: If using the PSM method, it should not be a direct regression. Add a test of balance, as well as a plot of kernel density before and after matching, and a plot of kernel density after matching. Response 5: Accepted. Based on your comment, we add a test of balance, as well as a plot of kernel density before and after matching. These additions enhance the reliability of the results obtained from the PSM test. Please see the Figure 3 on page 10 and the Figure 4 on page 11, lines 1751 to 1817, in the revised version for details.
Comments6: Conduct a robustness test using the Tobit model. Response 6: Accepted. In response to your opinion, we add Tobit model test with Mixed Logit and Random Effects Logit to enhance the robustness of benchmark regression results. Please see the last paragraph on page 8 and the Table 4 on page 9, lines 1575 to 1675, in the revised version for details.
Comments7: Please add between-group coefficient test to validate between-group differences of subgroups in the heterogeneity test. Response 7: Accepted. Following your comment, we add a between-group coefficient test using the Fisher's Permutation test method to enhance the reliability of the heterogeneity test results. This additional test allows us to directly compare differences between subgroups in the heterogeneity test. Please see the Table 7 on page 12 and the Table 8 on pages 13-14, lines 1982 to 2271, in the revised version for details.
Comments8: H2: The pilot policy of IP city promotes university innovation by enhancing universities' R&D investment. H3: The pilot policy of IP cities promotes university innovation by strengthening R&D cooperation. In the hypothesis section, the authors clearly use R&D investment as well as R&D cooperation as mediating mechanisms. Why does the mechanism analysis section use a triple difference model? This is incorrect. The authors either use a mediation model or rewrite the hypothesis section. Response 8: Accepted. Based on your opinion, we refresh the logic of the mechanism analysis and make the decision to not utilize the Triple Difference Model. Our mechanism tests essentially follow the principle of mediation effects, but because mediation mechanism has been seriously questioned in recent years, our mechanism analyses follow the principle of constructing recursive equations to be verified in three steps. Please see the last two paragraphs on page 14 and the first paragraphs on pages 15, lines 2273 to 2481, in the revised version for details.
Comments9: The triple difference model is essentially a derivative of the moderating effects model. Moderating effects models should include more than just interaction terms. It does not make economic sense to observe the significance of the coefficients of treat*time*M for the triple difference model only. Please refer to the Moderating Effects Model to modify the Triple Difference Model. Response 9: Accepted. In response to your comment, we review the literature on Triple Difference Model, and find that this model is not appropriate for our current research. This model is specifically designed to address the issue of biased estimates caused by the varying impact of other policies on pilot and non-pilot cities. We will better apply it in future related studies. Please see the last two paragraphs on page 14 and the first paragraphs on pages 15, lines 2273 to 2481, in the revised version for details.
|

Reviewer 2 Report
Comments and Suggestions for Authors
The topic of the paper is interesting but there are some problems needed to improve:
-Clearly state the research question and objectives. Make sure the objectives align with the research question, guiding the reader on what the study aims to achieve.
-Provide detailed information about the data sources, addressing any limitations or biases. Discuss the representativeness of the sample and consider employing additional robustness checks to test the sensitivity of the results.
-The study focuses on Chinese universities, and generalizing the findings to other countries or regions might not be straightforward. It's essential to discuss the external validity of the results.
-Establishing a causal relationship between IP policies and innovation can be challenging. It's crucial to acknowledge the possibility of correlations without causation and consider alternative explanations for the observed effects.
-Clearly define and justify the choice of variables. Discuss how each variable is measured and consider whether there are other relevant variables that could strengthen the analysis.
-Provide more specific and actionable policy implications based on the research findings. Consider how policymakers can use the results to inform decisions.
Comments on the Quality of English LanguageIt's ok.
Author Response
We appreciate the opportunity to revise our paper “Urban Intellectual Property Strategy and University Innovation: A Quasi-Natural Experiment from National Intellectual Property Model City”. Thank you very much for taking the time to review this manuscript. We have taken the suggestions to heart and believe we have successfully addressed all issues. Thus, we now upload a revised version of the manuscript that is now in much better shape after the process. You will also find a complete list of the detailed changes we have made in response to the comments on a point-by-point basis. We appreciate your willingness to review this version of our paper. Please let us know if you need any additional information from us.
Point-by-point response to Comments and Suggestions for Authors
|
Comments 1: Clearly state the research question and objectives. Make sure the objectives align with the research question, guiding the reader on what the study aims to achieve. |
|
Response 1: Accepted. Following your opinion, we directly state the research question in the first paragraph of the introduction; and broke down the research question and spell out the objectives of the paper in the second paragraph. The revised version is more reader-friendly. Please see the first two paragraphs on pages 1-2, lines 26 to 156 , in the revised version for details.
|
|
Comments 2: Provide detailed information about the data sources, addressing any limitations or biases. Discuss the representativeness of the sample and consider employing additional robustness checks to test the sensitivity of the results. |
|
Response 2: Accepted. Your opinions have been considered and we add Tobit model test with Mixed Logit and Random Effects Logit and the number of papers published by university as an explanatory variable to examine the robustness of our findings. The results from these models are consistent with benchmark regression results. Furthermore, we have provided detailed information about the data sources and data-processing procedures in the 3.3 Data description section of the article. And our study is based on a sample of all prefecture-level cities and undergraduate colleges and universities in China, making it representative for countries with similar development profiles to China. Please see the first paragraph on page 7, the last paragraph on page 8 and page 11, respectively, lines 1430 to 1452, 1562-1673 and 1822-1978 in turn, in the revised version for details.
Comments3: The study focuses on Chinese universities, and generalizing the findings to other countries or regions might not be straightforward. It's essential to discuss the external validity of the results. Response 3: Accepted. The primary purpose of the article's findings is to offer policy-level recommendations and insights on enhancing innovation in Chinese universities. These findings may also have some degree of universality and applicability for countries facing similar developmental circumstances as China. Much of the policy evaluation research that has been done also focuses on only one country (Xia, et al. 2023; Wu, et al. 2022; Hu, et al. 2021). In our future research, we plan to investigate the influence of intellectual property policies on university innovation in other countries, considering their specific contexts. List: [1] Xia, X.; Huang, T.; Zhang, S. The impact of intellectual property rights city policy on firm green innovation: A Quasi-Natural Experiment based on a Staggered DID Model. Systems 2023, 11, 209. [2] Wu, Z.; Fan, X.; Zhu, B.; Xia, J.; Zhang, L.; Wang, P. Do government subsidies improve innovation investment for new energy firms: A quasi-natural experiment of China's listed companies. Technological Forecasting and Social Change 2022, 175. [3] Hu, G.; Wang, X.; Wang, Y. Can the green credit policy stimulate green innovation in heavily polluting enterprises? Evidence from a quasi-natural experiment in China. Energy Economics 2021, 98, 3, 105134.
Comments4: Establishing a causal relationship between IP policies and innovation can be challenging. It's crucial to acknowledge the possibility of correlations without causation and consider alternative explanations for the observed effects. Response 4: Accepted. This study aims to assess the net effect of intellectual property policies on university innovation using DID method, and the objective is not to establish a causal relationship between IP policies and innovation. We explain why the DID method is suitable for our study in the 3.1 Model Setting section. The DID method helps reduce the bias caused by omitted variables and provides a more accurate estimate of policy effects. And existing studies have also used the DID model to evaluate the impact of intellectual property pilot cities on the quality of urban innovation and enterprise innovation, as well as the effects of other pilot policies, such as innovative pilot cities, on urban and enterprise innovation (Ji, et al. 2021; Xu , et al. 2021; Andrea, et al. 2018). List: [1] Ji, X.; Gu, N. Does the Establishment of Intellectual Property Model Cities Affect Innovation Quality?. Journal of Finance and Economics 2021, 5, 49-63. [2] Xu, Y.; Wei, D. Urban intellectual property strategy and enterprise innovation: a quasi-natural experiment from national intellectual property model city. Industrial Economics Research 2021, 4, 99-114. [3] Andrea, C.; Del, B. C. F. Smart innovative cities: The impact of Smart City policies on urban innovation. Technological Forecasting and Social Change 2018, 142, 373-383. Please see the penultimate paragraph on page 5, lines 1128 to 1138, in the revised version for details.
Comments5: Clearly define and justify the choice of variables. Discuss how each variable is measured and consider whether there are other relevant variables that could strengthen the analysis. Response 5: Accepted. Following your suggestion, we add in detail the literature references for variable selection and also provided clearer definitions for the variables. The measurements of the variables are primarily based on existing studies. Additionally, we also attempt to find other relevant variables that could strengthen the analysis. We identify the number of papers published by the university as the only one available variable, and used it as an explanatory variable to validate the reliability of the benchmark regression model. Please see the last three paragraphs on page 6 and the Table 1 on page 7, lines 1385 to 1420, in the revised version for details.
Comments6: Provide more specific and actionable policy implications based on the research findings. Consider how policymakers can use the results to inform decisions. Response 6: Accepted. In response to your opinion, we rethought the policy implications section and make it as specific and actionable as possible. Please see the first three paragraphs on page 16, lines 2689 to 2724, in the revised version for details.
|

Reviewer 3 Report
Comments and Suggestions for Authors
The paper is interesting and I learned something from it, I can't dispute that.
However, based on the available literature review in the manuscript, I cannot confirm that it is current. The cited literature is, first of all, insufficiently numerous - 29 literature references are by no means enough to provide a qualified explanation of the topicality and validity of the research, and secondly, the average age of the literary sources is, in my estimation, over 20 years (a literary source from 1959 is cited?!? as if this material were a chapter for a book and not for a journal).
Please, in my opinion, this is a serious omission, which points to insufficient knowledge of the subject matter in the field, and my remark follows this statement - the literature needs to be expanded and innovated, especially since the mentioned literature call has over five thousand citations.
Thank you.
Author Response
We appreciate the opportunity to revise our paper “Urban Intellectual Property Strategy and University Innovation: A Quasi-Natural Experiment from National Intellectual Property Model City”. Thank you very much for taking the time to review this manuscript. We have taken the suggestions to heart and believe we have successfully addressed all issues. Thus, we now upload a revised version of the manuscript that is now in much better shape after the process. You will also find a complete list of the detailed changes we have made in response to the comments. We appreciate your willingness to review this version of our paper. Please let us know if you need any additional information from us.
Response to Comments and Suggestions for Authors
|
Comments 1: The paper is interesting and I learned something from it, I can't dispute that. However, based on the available literature review in the manuscript, I cannot confirm that it is current. The cited literature is, first of all, insufficiently numerous - 29 literature references are by no means enough to provide a qualified explanation of the topicality and validity of the research, and secondly, the average age of the literary sources is, in my estimation, over 20 years (a literary source from 1959 is cited?!? as if this material were a chapter for a book and not for a journal).Please, in my opinion, this is a serious omission, which points to insufficient knowledge of the subject matter in the field, and my remark follows this statement - the literature needs to be expanded and innovated, especially since the mentioned literature call has over five thousand citations. |
|
Response 1: Accepted. In response to your feedback, we update and expand the literature references. To provide a more qualified explanation of the topicality and validity of the research, we add literature references to 54, with over half of them being from the last five years. We cite the classical literature because of its outstanding importance, for example, Nelson's paper (1959) was the first to propose the existence of externalities in basic research, Coase's paper (1960) first introduced the concept of property rights, and Solow's paper (1957) presented the first empirical evidence that economic development was driven by innovation mainly. List: [1] Nelson, R. R. The simple economics of basic scientific research. Journal of Political Economy 1959, 67, 297-306. [2] Coase, R. H. The problem of social cost. Journal of Law and Economics 1960, 3(1), 1-44. [3]Solow, R. M. Technical change and the aggregate production function. Review of Economics and Statistics 1957, 39, 312-320. Please see the references on pages 16-19, lines 2685 to 2853 , in the revised version for details.
|

Round 2
Reviewer 1 Report
Comments and Suggestions for Authors 1.Figure 2. IP pilot city policy placebo test is definitely wrong. Because the dashed line directly points towards 0 ,but the regression efficient is 0.6~0.7, So I suggest the author change this picture 2.The author could cite this article to strengthen the persuasiveness of the article Chen, M., Zhang, J., Xu, Z. et al. Does the setting of local government economic growth targets promote or hinder urban carbon emission performance? Evidence from China. Environ Sci Pollut Res 30, 117404–117434 (2023). https://doi.org/10.1007/s11356-023-30307-zAuthor Response
|
Comments 1: Figure 2. IP pilot city policy placebo test is definitely wrong. Because the dashed line directly points towards 0 ,but the regression efficient is 0.6~0.7, So I suggest the author change this picture. |
|
Response 1: Following your opinion, we carefully analyzed the contents of the subsection 4.2.3. Placebo Test. In Table 3. Benchmark regression estimates, the regression coefficients for the baseline regression Models (1), (2) and (3) are 0.699, 0.648 and 0.551, respectively. Model (1) excludes control variables, Model (2) incorporates university -level control variables, and model (3) includes both university-level and city-level control variables. To ensure the accuracy of the estimated results, the estimated coefficient presented in Figure 2. IP pilot city policy placebo test is derived from the regression coefficient of model (3) with the inclusion of university-level and city-level control variables. Therefore, the regression efficient in Figure 2. IP pilot city policy placebo test is 0. 551, and not between 0.6 and 0. 7.
|
|
Comments 2: The author could cite this article to strengthen the persuasiveness of the article Chen, M., Zhang, J., Xu, Z. et al. Does the setting of local government economic growth targets promote or hinder urban carbon emission performance? Evidence from China. Environ Sci Pollut Res 30, 117404–117434 (2023). https: //doi.org/ 10.1007/ s11356-023-30307-z. |
|
Response 2: Accepted. We have included this paper in our references. Thank you for recommending it! After studying this paper, we found that its theme, structure, research design, and methodology were truly inspiring us and greatly assisted us in advancing our research, such as to explore whether setting local economic growth targets can drive improvements in university innovation. Please see the page 19, lines 727 to 729 , in the revised version for details. |

Reviewer 2 Report
Comments and Suggestions for Authors
I think the authors made good job improving paper.
Comments on the Quality of English LanguageIt's ok.
Author Response
|
Comments: I think the authors made good job improving paper. |
|
Response : We would like to express our gratitude for the opportunity to revise our paper titled 'Urban Intellectual Property Strategy and University Innovation: A Quasi-Natural Experiment from National Intellectual Property Model City.' We sincerely appreciate the time and effort you have dedicated to reviewing this manuscript once again. Your approval serves as a great source of encouragement for us. There may be some limitations in our study, but we assure that we will make continuous efforts to address and expand upon them in future research. |

Reviewer 3 Report
Comments and Suggestions for Authors
Dear colleagues,
quoting literary classics is perfectly fine when you are writing a book or a college textbook, so you want to show off your knowledge.
Journals are different - they have a different function and a different readership; classics are not interesting.
Author Response
|
Comments: Dear colleagues, quoting literary classics is perfectly fine when you are writing a book or a college textbook, so you want to show off your knowledge. Journals are different - they have a different function and a different readership; classics are not interesting. |
|
Response : Accepted. Following your opinion, we have essentially removed all of the classics literature, and more than half of the references cited in the revised version are the most recent research from the last five years. Thank you very much for your guidance, and we will keep this in mind in our future research. |
Please see the References on pages, in the revised version for details.
